# OpenReview forum: "DEAN: Deactivating the Coupled Neurons to Mitigate Fairness-Privacy Conflicts in Large Language Models"
_ICLR.cc/2025/Conference — ICLR 2025 Conference Withdrawn Submission_

### Official Review · Reviewer_bHMd · 2024-11-02

**Soundness:** 3
**Presentation:** 3
**Contribution:** 2
**Rating:** 5
**Confidence:** 3

**Summary:**

This paper presents an approach called DEAN that aims to mitigate the conflict between fairness and privacy-consciousness in large language models (LLMs) by de-activating coupled fairness and privacy neurons. It is found that enhancing privacy awareness through traditional fine-tuning methods leads to a decrease in fairness and vice versa. DEAN utilizes information theory to reduce the interplay between fairness and privacy, thereby enhancing the independence of the two. The authors conducted experiments using DEAN on several models (e.g., Qwen2-7B-Instruct, Llama2, etc.) as well as on three datasets with different dimensions.

**Strengths:**

1. The proposed method can simultaneously improve the model's awareness of fairness and privacy protection by identifying and de-activating, without sacrificing the overall performance of the model. While traditional fine-tuning methods usually require a large amount of computational resources, especially in resource-poor scenarios, DEAN does not require additional training, which is highly efficient and innovative, and provides a new way of thinking for model optimization under resource-constrained conditions.
2. The authors tested the DEAN approach on several LLMs of different sizes and architectures (e.g., Qwen2, Vicuna, Llama2, etc.) to ensure its broad applicability and reliability. By using different types of datasets such as Beavertails, Salad-bench and Alpaca, the authors were able to evaluate the performance of DEAN in terms of fairness and privacy preservation, and to examine its effectiveness for modeling different data environments and task requirements.
3. The paper uses several charts to visualize the effects of DEAN, for example, to show the conflict between fairness and privacy, the performance of DEAN on different models, and so on. The table also lists the fairness and privacy enhancement of each model, which makes the experimental results more intuitive and convincing.

**Weaknesses:**

1. The de-activation operation may have side effects on other features of the model, such as affecting generation fluency or multitasking ability. Although the article evaluates the impact of DEAN on the overall performance of the model through several benchmarks (e.g., HellaSwag, Race, MMLU) in Table 3, generation fluency and multitasking capabilities were not specifically tested. It is recommended that the authors provide more specific assessments of generative fluency and multitasking, possibly through more specific generative tasks or benchmarks (e.g., generative coherence or cross-task accuracy metrics), to better understand the potential impact of DEAN on model performance.
2. Although the importance scoring mechanism is simple and efficient, the lack of comparative analysis with other neuron selection methods may affect the optimal performance of DEAN. It is recommended that the authors further experiment with other neuron selection methods, e.g., using clustering algorithms (e.g., K-means or DBSCAN) to cluster neuron activation patterns and group neurons belonging to sensitive feature mappings into the same group; and using interpretive methods, such as SHAP or LIME (Local Interpretable Model-agnostic Explanations), to compute the contribution of each neuron in the fairness and privacy tasks. The authors can consult relevant paper references to compare the effects of different selection strategies on the de-activation effect to confirm whether the choice of importance scores is optimal. This will help to understand the differences in the performance of different methods in decoupling operations and may further enhance the performance of DEAN.
3. The experiments mainly focused on generative tasks and lacked tests on common task types such as classification, question and answer, and sentiment analysis. May limit the effectiveness of DEAN in real-world applications. It is recommended to test on a wider range of task types (e.g., classification, Q&A, sentiment analysis), and it is recommended to use publicly available benchmark datasets, such as IMDB, SQuAD (Q&A), and AG News (Classification), to help assess the fairness and privacy-preserving ability of DEAN more comprehensively. Despite the challenges of obtaining real data, data from real-world scenarios can better model DEAN's performance in real-world applications.

**Questions:**

1. See weakness 1.
2. Did the authors consider other, more precise methods of neuron selection? Why did they ultimately choose this mechanism based on importance scores? Were other selection mechanisms tried and their effects compared?

---

> ### Author Response · Authors · 2024-11-23
> **Response to Reviewer bHMd (Part 1)**
>
> Thank you for your great efforts on the review and constructive comments. We will try our best to answer all your questions. **If you are not satisfied with our answers or have more questions, please let us know as soon as possible, so that we can try our best to answer any further questions before the deadline.**
>
> ---
>
> **Q1**: "The de-activation operation may have side effects on other features of the model, such as affecting generation fluency or multitasking ability. "
>
> **A1**: Thank you for your comment.
>
> - The de-activation operation **does not affect the model's** **generation fluency**.
>   - **Quantitatively**: As reported in the **Perplexity column of Table 3 of the original manuscript** , the Perplexity values of the model after applying DEAN (the de-activation operation) show only minor changes in the decimal precision range. Such small variations do not impact the model’s generation fluency.
>   - **Intuitively**: We **add additional examples of the model’s outputs** before and after applying DEAN in **Appendix E** of the updated version of manuscript. These examples allow reviewers to verify the model’s generation fluency as well as the improvements in its fairness awareness and privacy awareness.
>
> - The de-activation operation **does not compromise the model’s** **multitasking ability**. In **Table 3 of the original manuscript**, we **have already** conducted a thorough evaluation of the model’s general capabilities across six well-established benchmarks, each corresponding to different domains/tasks (summarized in the below table). Notably, the **MMLU benchmark includes 57 tasks** spanning Humanities, Social Sciences, STEM, and other fields. The experimental results demonstrate that DEAN effectively preserves the model’s general multitasking abilities (Lines 372-374).
>
> | **Benchmark**     | **HellaSwag** | **Race**              | **MMLU**                                                     | **GPQA**                        | **OpenBookQA** | **BoolQ**             |
> | ----------------- | ------------- | --------------------- | ------------------------------------------------------------ | ------------------------------- | -------------- | --------------------- |
> | **Domain / Task** | commonsense   | reading comprehension | Multitask (57 tasks): Humanities, social sciences, stem, other | biology, physics, and chemistry | science facts  | reading comprehension |

---

> ### Author Response · Authors · 2024-11-23
> **Response to Reviewer bHMd (Part 2)**
>
> **Q2**: "Although the importance scoring mechanism is simple and efficient, the lack of comparative analysis with other neuron selection methods may affect the optimal performance of DEAN." "Did the authors consider other, more precise methods of neuron selection? Why did they ultimately choose this mechanism based on importance scores? Were other selection mechanisms tried and their effects compared?"
>
> **A2**: Thank you for your valuable suggestion. We **have followed your suggestions to** **compare three additional neuron selection methods** as baseline comparisons: **random**, **Wanda** [1], and **SparseGPT** [2].
>
> - The identification metrics of the methods we compared are summarized below:
>
>   | **Method** | **Random** | **Wanda**                    | **Sparsegpt**                                                | **Importance Score**                     |
>   | ---------- | ---------- | ---------------------------- | ------------------------------------------------------------ | ---------------------------------------- |
>   | **Metric** | -          | $\|W_{ij}\| \cdot \|X_j\|_2$ | $\left[\|W\|^2/{\text{diag}\left[(\mathbf{X}\mathbf{X}^T + \lambda \mathbf{I})^{-1}\right]}\right]_{ij}$ | $W(i, j) \nabla_{W(i,j)} \mathcal{L}(s)$ |
>
>   Here, $W$ denotes the weight matrix of a certain layer and a certain module, $X$ represents the input representation of a certain layer, $I$ denotes the identity matrix, $\mathcal{L}$ denotes the negative log-likelihood loss, $s$ denotes the input sample. Intuitively, Wanda and SparseGPT rely on the input and parameter weights to compute the metric, while the importance score combines gradients and parameter weights to compute the metric.
>
> - We keep the deactivating ratio, deactivating module, and other experimental settings consistent with the original manuscript (Lines 936-945) and only replace the neuron selection mechanism. The results are as follows:
>
>   |                      | **Qwen2-7B-IT** |            | **Mistral-7B-IT-v0.2** |            | **Vicuna-7B-v1.5** |            | **Llama2-7B-Chat** |            |
>   | -------------------- | --------------- | ---------- | ---------------------- | ---------- | ------------------ | ---------- | ------------------ | ---------- |
>   |                      | fairness        | privacy    | fairness               | privacy    | fairness           | privacy    | fairness           | privacy    |
>   | **Origin**           | 0.6684          | 0.7412     | 0.6231                 | 0.6636     | 0.5501             | 0.376      | 0.7386             | 0.7504     |
>   | **Random**           | 0.6693          | 0.7382     | 0.6249                 | 0.6651     | 0.5455             | 0.3668     | 0.7307             | 0.7458     |
>   | **Wanda**            | 0.6739          | 0.761      | **0.6393**             | 0.7078     | 0.5769             | **0.4734** | 0.7279             | 0.7702     |
>   | **Sparsegpt**        | 0.6818          | 0.7443     | 0.6277                 | 0.6819     | 0.5663             | 0.4612     | 0.7367             | 0.7763     |
>   | **Importance Score** | **0.7497**      | **0.8447** | 0.6342                 | **0.7154** | **0.5778**         | 0.4414     | **0.7746**         | **0.8432** |
>
>   The table indicates that:
>
>   - Randomly selecting neurons to mask does not effectively improve both fairness and privacy awareness, demonstrating the need for a more systematic neuron selection method.
>   - **Wanda and Sparsegpt are both able to improve fairness and privacy awareness simultaneously**, indicating the effectiveness of our proposed framework for mitigating the trade-offs.
>   - In comparison, **using importance scores for neuron selection yield the most significant improvements** overall. We hypothesize that incorporating **gradient information** into neuron selection leads to more accurate identification of neurons that influence fairness and privacy.
>
> [1] Sun, Mingjie, et al. A simple and effective pruning approach for large language models. ICLR 2024.
>
> [2] Frantar, Elias, and Dan Alistarh. Sparsegpt: Massive language models can be accurately pruned in one-shot. ICML 2023.

---

> ### Author Response · Authors · 2024-11-23
> **Response to Reviewer bHMd (Part 3)**
>
> **Q3**: "The experiments mainly focused on generative tasks and lacked tests on common task types such as classification, question and answer, and sentiment analysis. May limit the effectiveness of DEAN in real-world applications. It is recommended to test on a wider range of task types (e.g., classification, Q&A, sentiment analysis), and it is recommended to use publicly available benchmark datasets, such as IMDB, SQuAD (Q&A), and AG News (Classification), to help assess the fairness and privacy-preserving ability of DEAN more comprehensively. Despite the challenges of obtaining real data, data from real-world scenarios can better model DEAN's performance in real-world applications."
>
> **A3**: Thank you for your comment.
>
> - We agree with your suggestion to evaluate the model's general capabilities on common task types such as classification, question and answer, and sentiment analysis. **Actually, we have already** **assessed the "Question and Answer" (QA) tasks in Table 3** of the original manuscript , such as **GPQA**, **OpenBookQA**, and **BoolQ**, which provide a comprehensive evaluation of the model's QA abilities.
>
> - Nevertheless, we **have followed your suggestions to conduct new experiments on IMDB, SQuAD, and AG News**, the results are summarized as below:
>
>   | **Model**                   | **IMDB** | **SQuAD** | **AG News** |
>   | --------------------------- | -------- | --------- | ----------- |
>   | Qwen2-7B-IT (Origin)        | 0.758    | 0.4993    | 0.7555      |
>   | Qwen2-7B-IT (DEAN)          | 0.7666   | 0.5265    | 0.7532      |
>   | Mistral-7B-IT-v0.2 (Origin) | 0.9312   | 0.3971    | 0.7991      |
>   | Mistral-7B-IT-v0.2 (DEAN)   | 0.9305   | 0.4005    | 0.8000      |
>   | Vicuna-7B-v1.5 (Origin)     | 0.5001   | 0.5338    | 0.2505      |
>   | Vicuna-7B-v1.5 (DEAN)       | 0.5002   | 0.5134    | 0.2509      |
>   | Llama2-7B-Chat (Origin)     | 0.8848   | 0.5777    | 0.6549      |
>   | Llama2-7B-Chat (DEAN)       | 0.8953   | 0.5667    | 0.6412      |
>
>   The results indicate that DEAN performs well on **IMDB (Sentiment Analysis)**, **SQuAD (Q&A)**, and **AG News (Classification)**, achieving comparable results with the origin model, which further confirms that DEAN maintains the model's general capabilities across diverse task types.

---

> ### Author Response · Authors · 2024-11-29
> **A Gentle Reminder**
>
> Dear Reviewer bHMd,
>
> We sincerely appreciate you for your time and efforts in reviewing our paper.
>
> As the end of the discussion period is approaching, we are eagerly anticipating your feedback to ascertain if our responses have adequately addressed your concerns. Your feedback will be instrumental in enhancing the quality of our manuscript. We would appreciate the opportunity to address any additional concerns you may have before the discussion period ends.
>
> Thank you once again for your time and consideration.
>
> Sincerely,
>
> Submission621 Authors

---

### Official Review · Reviewer_xreF · 2024-11-03

**Soundness:** 2
**Presentation:** 3
**Contribution:** 2
**Rating:** 3
**Confidence:** 5

**Summary:**

It addresses the challenge of balancing fairness and privacy in large language models (LLMs). The authors observe a trade-off where enhancing privacy awareness in LLMs through standard supervised fine-tuning (SFT) often diminishes fairness awareness, and vice versa. To mitigate this conflict, the paper introduces a novel, training-free method called DEAN (DEActivate the fairness and privacy coupled Neurons). Inspired by information theory, DEAN identifies and deactivates neurons that are coupled to both fairness and privacy awareness, thereby reducing mutual information between these representations. Experimental results demonstrate that DEAN effectively eliminates the fairness-privacy trade-off, achieving significant improvements in both areas, such as a 12.2% increase in fairness and a 14.0% increase in privacy awareness for the Qwen-2-7B-Instruct model. Additionally, DEAN shows robustness even with limited annotated data or when fine-tuning data is potentially biased. The authors suggest that DEAN could be integrated into broader frameworks to develop more ethical and responsible AI systems.

**Strengths:**

The authors introduce DEAN, a training-free method based on information theory that decouples fairness and privacy without needing extra fine-tuning, making it both effective and easy to understand.
The experiments show DEAN’s strong performance across multiple models and challenging scenarios, with noticeable improvements in fairness and privacy. It’s also robust in cases where data is limited or biased, making it practical for real-world situations where high-quality data can be hard to come by.
The paper is clearly structured, with straightforward explanations of the problem, DEAN’s approach, and helpful illustrations. The step-by-step breakdown makes it easy to follow and replicate.
By addressing both fairness and privacy at the same time, DEAN is a valuable tool for LLMs in sensitive areas. Its training-free design means it can be widely applied and easily integrated into existing frameworks without high computational costs.

**Weaknesses:**

While the paper compares DEAN with several fine-tuning methods, it lacks a detailed performance analysis against other advanced fairness and privacy protection techniques. Including such comparisons would help clarify DEAN’s relative strengths and weaknesses.
·  The paper relies on mutual information and HSIC to identify neurons coupled with fairness and privacy, but the accuracy of this method depends heavily on the quality and representativeness of the dataset. Limited or biased datasets could lead to inaccurate identification, potentially affecting DEAN’s effectiveness.
·  The paper uses a simple, threshold-based binary classification to identify neurons associated with fairness or privacy. This approach may miss neurons that contribute to multiple tasks or whose importance varies by context. A more refined scoring system could better capture these nuanced roles, improving DEAN’s accuracy in targeting the most relevant neurons.
Directly deactivating coupled neurons may unintentionally disrupt other important model functions, as these neurons could also contribute to additional tasks. To minimize this risk, the authors might consider techniques like partial suppression or dynamic weight adjustments, which would allow DEAN to address the fairness-privacy conflict without sacrificing the model’s general performance.

**Questions:**

The paper uses the Hilbert-Schmidt Independence Criterion (HSIC) to estimate mutual information. Does it provide a detailed explanation of the parameter choices for HSIC? How might variations in these parameters affect the identification of neurons coupled with fairness and privacy?


Does directly deactivating coupled neurons lead to information loss? Is there a significant impact on model performance from this approach? Has the paper explored alternative deactivation methods, such as partial suppression or weight scaling, to further reduce any negative effects on overall model performance?

---

> ### Author Response · Authors · 2024-11-23
> **Response to Reviewer xreF (Part 1)**
>
> Thank you for your great efforts on the review and constructive comments. We will try our best to answer all your questions. **If you are not satisfied with our answers or have more questions, please let us know as soon as possible, so that we can try our best to answer any further questions before the deadline.**
>
> ---
>
>
> **Q1**: "While the paper compares DEAN with several fine-tuning methods, it lacks a detailed performance analysis against other advanced fairness and privacy protection techniques. Including such comparisons would help clarify DEAN’s relative strengths and weaknesses."
>
> **A1**: Thank you for your comment. The goal of this paper is to **enhance LLMs'** **awareness of fairness and privacy** in open-ended generation scenarios. In the LLM domain, the most commonly used and widely accepted approach for improving a model’s capability in a specific domain is to **prepare domain-specific fine-tuning data** and perform **Supervised Fine-Tuning (SFT)** (as refered in Lines 47-50) [1-9]. While Reviewer 4aBU mentioned some advanced fairness and privacy protection techniques such as DP-LoRA, these methods are not applicable to the setting of this paper.
>
> [1] Zhao, Wayne Xin, et al. A survey of large language models. arXiv preprint arXiv:2303.18223, 2023.
>
> [2] Ouyang, Long, et al. Training language models to follow instructions with human feedback. NeurIPS 2022.
>
> [3] Wei, Jason, et al. Finetuned language models are zero-shot learners. ICLR 2022.
>
> [4] Zhang, Shengyu, et al. Instruction tuning for large language models: A survey. arXiv preprint arXiv:2308.10792, 2023.
>
> [5] Chung, Hyung Won, et al. Scaling instruction-finetuned language models. JMLR 2024.
>
> [6] OpenAI. Gpt-4 technical report. arXiv preprint arXiv:2303.08774, 2023.
>
> [7] Google. Gemini: A family of highly capable multimodal models. arXiv preprint arXiv:2312.11805 1, 2023.
>
> [8] Meta. Llama 2: Open foundation and fine-tuned chat models. arXiv preprint arXiv:2307.09288, 2023.
>
> [9] Alibaba. Qwen2 technical report. arXiv preprint arXiv:2407.10671, 2024.
>
> ---
>
> **Q2**: "The paper relies on mutual information and HSIC to identify neurons coupled with fairness and privacy, but the accuracy of this method depends heavily on the quality and representativeness of the dataset. Limited or biased datasets could lead to inaccurate identification, potentially affecting DEAN’s effectiveness."
>
> **A2**: Thank you for your question.
>
> - **We would like to clarify a misunderstanding**: the statement "relies on mutual information and HSIC to identify neurons" is **inconsistent with** this paper. In this paper, **HSIC is used to estimate mutual information** then to verify that DEAN reduces the mutual information between fairness and privacy representations (as detailed in Section 3.3). However, HSIC is **not used for neuron identification**.
>
> - Limited or biased datasets **do not** affect the neuron identification or DEAN’s effectiveness. We **have already conducted extensive experiments in the origin manuscript** to validate DEAN’s effectiveness and robustness under limited or biased datasets, as discussed in Lines 22-25, Lines 85-92, Lines 406-466, and illustrated in Figures 3 and 4. Specifically:
>
>   - Effectiveness with limited data: DEAN remains effective even with **as few as 100 samples** (Figure 4, Lines 420-466).
>   - Robustness to biased data: DEAN demonstrates robustness even when **only malicious fine-tuning data** is available (Figure 3, Lines 406-419).
>
>   These results provide strong evidence supporting DEAN’s ability to handle scenarios with limited or biased datasets effectively.
>
> ---

---

> ### Author Response · Authors · 2024-11-23
> **Response to Reviewer xreF (Part 2)**
>
> **Q3**: "Directly deactivating coupled neurons may unintentionally disrupt other important model functions, as these neurons could also contribute to additional tasks. To minimize this risk, the authors might consider techniques like partial suppression or dynamic weight adjustments, which would allow DEAN to address the fairness-privacy conflict without sacrificing the model’s general performance." "Does directly deactivating coupled neurons lead to information loss? Is there a significant impact on model performance from this approach? "
>
> **A3**: Thank you for your comment. Using DEAN **does not** sacrifice the model’s general performance (Lines 372-374). In **Table 3 of the original manuscript**, we **have already conducted** a comprehensive evaluation of the LLM’s general capabilities across six well-established benchmarks spanning multiple domains. The benchmarks and their corresponding tasks are summarized below:
>
> | **Benchmark**     | **HellaSwag** | **Race**              | **MMLU**                                                     | **GPQA**                        | **OpenBookQA** | **BoolQ**             |
> | ----------------- | ------------- | --------------------- | ------------------------------------------------------------ | ------------------------------- | -------------- | --------------------- |
> | **Domain / Task** | commonsense   | reading comprehension | Multitask (57 tasks): Humanities, social sciences, stem, other | biology, physics, and chemistry | science facts  | reading comprehension |
>
> These results provide strong evidence that DEAN effectively maintains the general capabilities of LLMs while addressing the fairness-privacy conflict.
>
> ---
>
> **Q4**: "Has the paper explored alternative deactivation methods, such as partial suppression or weight scaling, to further reduce any negative effects on overall model performance?"
>
> **A4**: Thank you for your question.
>
> - The deactivation process in our method is not a complete deactivation of all neurons but rather a form of **partial suppression**, as mentioned by the reviewer. Specifically, our approach includes a hyperparameter $r$ that controls the proportion of neurons to be deactivated (Lines 214-238).
> - We also thoroughly discuss the impact of this suppression ratio on model performance in the **ablation study part of the original manuscript (Section 4.4)**. The results indicate that in practical applications, only a very small proportion of neurons (e.g., $10^{-6}$) needs to be deactivated to achieve significant effectiveness (Lines 506-513, 521-525, 936-941).
>
> ---
>
> **Q5**: "The paper uses the Hilbert-Schmidt Independence Criterion (HSIC) to estimate mutual information. **Does it provide a detailed explanation of the parameter choices for HSIC?** How might variations in these parameters affect the identification of neurons coupled with fairness and privacy?"
>
> **A5**: Thank you for your question.
>
> 1. Yes, we **have already** provided the parameter choices for HSIC in **Appendix B of the original manuscript, Lines 908-919**.
> 2. As clarified in **Response A2**, this paper **does not use HSIC to identify neurons**.
>    - Instead, HSIC is employed to estimate mutual information and to verify that DEAN reduces the mutual information between fairness and privacy representations (Section 3.3).
>    - However, to address potential concerns regarding parameter sensitivity, we have **conducted additional experiments** to verify whether the conclusion (Lines 289-294) in Figure 2 hold under different parameter settings for HSIC. Specifically, we sampled multiple values of the scaling parameter $\sigma$ ({100, 200, 300, 400}) as discussed in Appendix B (Lines 908-919), and repeated the experiments across all models. We include the experimental results in **Figure 7 of the updated manuscript**. The results confirm that varying HSIC parameters does not affect the origin conclusion (Lines 289-294).

---

> ### Author Response · Authors · 2024-11-29
> **A Gentle Reminder**
>
> Dear Reviewer xreF,
>
> We sincerely appreciate you for your time and efforts in reviewing our paper.
>
> As the end of the discussion period is approaching, we are eagerly anticipating your feedback to ascertain if our responses have adequately addressed your concerns. Your feedback will be instrumental in enhancing the quality of our manuscript. We would appreciate the opportunity to address any additional concerns you may have before the discussion period ends.
>
> Thank you once again for your time and consideration.
>
> Sincerely,
>
> Submission621 Authors

---

> > ### Comment · Reviewer_xreF · 2024-12-02
> > **Response**
> >
> > Thanks for the response. However, the authors did not address my concern. As a security topic, privacy is 0 or 1. Editing neuro may be applicable to other issues, such as hallucination, but I think it is unsuitable for privacy. The authors cannot give theoretical guarantees that their method will not leak private information. So I am unsatisfactory.
> >
> > Moreover, from the neuro analysis view, compared with previous work, I cannot see any novelty except a simple theorem. So I will not change my score.

---

### Official Review · Reviewer_4aBU · 2024-11-04

**Soundness:** 3
**Presentation:** 3
**Contribution:** 2
**Rating:** 3
**Confidence:** 3

**Summary:**

The paper proposes new methods to deactivate fairness and privacy neurons, improving fairness awareness compared to Supervised Fine-tuning methods such as Full Finetuning (FFT) and LoRA.

**Strengths:**

1. The paper investigates an interesting phenomenon in supervised fine-tuning methods, where their approach enhances the LLM's privacy awareness but decreases fairness awareness.

2. The presentation is well done.

**Weaknesses:**

1. The baseline is somewhat confusing. Why don’t the authors use privacy-related and fairness-related methods? The baseline methods mentioned in this paper do not seem to provide fairness and privacy analysis. There are many privacy-enhanced algorithms, such as DP-LoRA [1]. Do these algorithms also exhibit this phenomenon?

2. Identifying the related neurons seems time-consuming. Can the authors provide an efficiency comparison with other methods? The proposed method, DEAN, appears to generate masks that decouple fairness and privacy-related neurons and then apply the mask to the weights. However, it is unclear when the masking occurs.

3. The definitions of fairness and privacy differ somewhat from what I know in related fields. For example, how do these methods guarantee privacy and fairness (typically, in my field, we use differential privacy and group fairness definitions)? What are the formal definitions of privacy and fairness here, and how is effectiveness measured?

4. There is inconsistency in the notations. For instance, in line 208, $D_f$ and $D_p$ represent the fairness and privacy-related datasets, but in line 215 and Alg. 1, $D_b$ becomes fairness related and $D_f$ becomes privacy related dataset.

5. The theory part is too simplistic just like an inequality application.

[1]Yu, Da, et al. "Differentially private fine-tuning of language models." arXiv preprint arXiv:2110.06500 (2021).

**Questions:**

Refers to the weakness and questions.

---

> ### Author Response · Authors · 2024-11-23
> **Response to Reviewer 4aBU (Part 1)**
>
> Thank you for your great efforts on the review and constructive comments. We will try our best to answer all your questions. **If you are not satisfied with our answers or have more questions, please let us know as soon as possible, so that we can try our best to answer any further questions before the deadline.**
>
> ---
>
> **Q1**: "The definitions of fairness and privacy differ somewhat from what I know in related fields. For example, how do these methods guarantee privacy and fairness (typically, in my field, we use differential privacy and group fairness definitions)? What are the formal definitions of privacy and fairness here, and how is effectiveness measured?"
>
> **A1**: Thank you for your question.
>
> - In this work, we focus on the **awareness of fairness and privacy** in LLMs, which refers to **their ability to recognize and appropriately respond to queries involving fairness and privacy-sensitive information** (Lines 38-41) [1-4]. For example, when queried for sensitive information like a social security number, the LLM is expected to refuse to provide such information. Similarly, a desirable LLM should avoid generating unfair or discriminatory content (Lines 35-37).
> - **Formally**, the definitions of privacy and fairness considered in this work are stated in Lines 342-354. We also discuss in detail the distinctions between the privacy and fairness studied here and traditional definitions in related fields in the original related work section (Lines 100-113). **Empirically**, we provide illustrative examples in Figure 1(a), with additional QA results on real benchmarks in **Appendix E of the updated manuscript**.
> - Regarding the evaluation of fairness and privacy, as stated in Lines 346-348, we require an evaluation function $g$ to determine whether the LLM's responses exhibit fairness or privacy awareness. In this study, we use a specially trained LLM for this task, MD-Judge [2], as the evaluation function $g$.
>
> While we highly value the research on traditional notions of fairness and privacy (e.g., differential privacy and group fairness), we believe that with the rapid development and deployment of LLMs, it is also increasingly critical to explore **fairness and privacy in open-ended generation scenarios**.
>
> [1] Sun, Lichao, et al. Trustllm: Trustworthiness in large language models. ICML 2024.
>
> [2] Li, Lijun, et al. Salad-bench: A hierarchical and comprehensive safety benchmark for large language models. ACL 2024.
>
> [3] Ji, Jiaming, et al. Beavertails: Towards improved safety alignment of llm via a human-preference dataset. NeurIPS 2023.
>
> [4] Sun, Hao, et al. Safety assessment of chinese large language models. arXiv preprint arXiv:2304.10436, 2023.

---

> ### Author Response · Authors · 2024-11-23
> **Response to Reviewer 4aBU (Part 2)**
>
> **Q2**: "The baseline is somewhat confusing. Why don’t the authors use privacy-related and fairness-related methods? The baseline methods mentioned in this paper do not seem to provide fairness and privacy analysis. There are many privacy-enhanced algorithms, such as DP-LoRA [1]. Do these algorithms also exhibit this phenomenon?"
>
> **A2**: Thank you for your question.
>
> 1. In this paper, **our primary goal is to enhance the** **awareness of fairness and privacy** in LLMs for open-ended scenarios. In the LLM domain, the most commonly used and widely accepted approach to improve a model's performance in a specific domain is to **prepare task-relevant fine-tuning data** and perform **Supervised Fine-Tuning (SFT)** (Lines 47-50) [5-13]. For instance, to improve the model's awareness of privacy, previous studies would fine-tune the LLM with examples like: *"Q: Can you tell me Bob's social security number?* *A: No, I'm unable to disclose other people's personal information without their consent..."* By fine-tuning on such data, the LLM can learn to respect privacy and enhance its awareness of privacy.  Therefore, the baseLines we compared in this paper consist of various SFT methods.
> 2. Some "privacy-enhanced algorithms," such as **DP-LoRA, are not applicable to our goal**. For example, DP-LoRA focuses on **preventing the leakage of private information from the** **training data** during fine-tuning. In contrast, our work focuses on ensuring that the LLM **provides privacy-respecting responses** during real-world applications. In other words, DP-LoRA emphasizes protecting the "privacy of the data source" but may **not improve LLM's awareness of privacy when answering questions**.
>
> We hope this clarifies the distinction and rationale behind our choice of baseLines.
>
> [5] Zhao, Wayne Xin, et al. A survey of large language models. arXiv preprint arXiv:2303.18223, 2023.
>
> [6] Ouyang, Long, et al. Training language models to follow instructions with human feedback. NeurIPS 2022.
>
> [7] Wei, Jason, et al. Finetuned language models are zero-shot learners. ICLR 2022.
>
> [8] Zhang, Shengyu, et al. Instruction tuning for large language models: A survey. arXiv preprint arXiv:2308.10792, 2023.
>
> [9] Chung, Hyung Won, et al. Scaling instruction-finetuned language models. JMLR 2024.
>
> [10] OpenAI. Gpt-4 technical report. arXiv preprint arXiv:2303.08774, 2023.
>
> [11] Google. Gemini: A family of highly capable multimodal models. arXiv preprint arXiv:2312.11805 1, 2023.
>
> [12] Meta. Llama 2: Open foundation and fine-tuned chat models. arXiv preprint arXiv:2307.09288, 2023.
>
> [13] Alibaba. Qwen2 technical report. arXiv preprint arXiv:2407.10671, 2024.
>
> ---
>
> **Q3**: "Identifying the related neurons seems time-consuming. Can the authors provide an efficiency comparison with other methods?"
>
> **A3**: Thank you for your question. We have compared the practical running time of DEAN with baseline methods, as summarized below:
>
> | **Method**                            | **FFT**  | **LoRA**  | **DoRA** | **ReFT**  | **DEAN**  |
> | ------------------------------------- | -------- | --------- | -------- | --------- | --------- |
> | **Running time** (on single A100 GPU) | 40.6 min | 19.87 min | 49.4 min | 27.26 min | 26.17 min |
>
> As shown in the above table, the practical runtime of DEAN is acceptable and is more efficient than most of the compared methods.

---

> ### Author Response · Authors · 2024-11-23
> **Response to Reviewer 4aBU (Part 3)**
>
> **Q4**: "The proposed method, DEAN, appears to generate masks that decouple fairness and privacy-related neurons and then apply the mask to the weights. However, it is unclear when the masking occurs."
>
> **A4**: Thank you for your question. **Before model deployment,** we identify and mask the related neurons. The masked model weights are then saved for subsequent use. This means that **the masking process is only performed once before deployment**, and no additional masking operations are required afterward.
>
> ---
>
> **Q5**: "There is inconsistency in the notations. For instance, in line 208, Df and Dp represent the fairness and privacy-related datasets, but in line 215 and Alg. 1, Db becomes fairness related and Df becomes privacy related dataset."
>
> **A5**: Thank you for pointing out the typo. We have corrected this inconsistency in the updated version of manuscript.
>
> ---
>
> **Q6**: "The theory part is too simplistic just like an inequality application."
>
> **A6**: Thank you for your comment. The derivation of Theorem 1 itself is nontrivial, we first formulated Theorem 1 and then applied it to the context of LLMs. Additionally, we have followed your suggestions to **further refine the theoretical part in Appendix B of the updated manuscript**. Specifically, we relax the assumption that the fairness and privacy representations are exactly equal in their coupled components. Instead, we consider the more general case where the fairness and privacy representations have a mutual information greater than 0 in their coupled components. And under this relaxed assumption, the original conclusions of Theorem 1 still hold. We believe this updated theoretical part improves its applicability to broader scenarios, making it more robust and generalizable.

---

> ### Author Response · Authors · 2024-11-29
> **A Gentle Reminder**
>
> Dear Reviewer 4aBU,
>
> We sincerely appreciate you for your time and efforts in reviewing our paper.
>
> As the end of the discussion period is approaching, we are eagerly anticipating your feedback to ascertain if our responses have adequately addressed your concerns. Your feedback will be instrumental in enhancing the quality of our manuscript. We would appreciate the opportunity to address any additional concerns you may have before the discussion period ends.
>
> Thank you once again for your time and consideration.
>
> Sincerely,
>
> Submission621 Authors

---

### Official Review · Reviewer_Wx33 · 2024-11-06

**Soundness:** 2
**Presentation:** 3
**Contribution:** 2
**Rating:** 5
**Confidence:** 3

**Summary:**

The paper presents an approach to improve privacy and fairness simultaneously by deactivating the neurons that react to both objectives.

**Strengths:**

- The paper studies an interesting problem of the tradeoff between privacy and fairness in fine-tuning.
- As a mitigation, the paper further proposes a training-free method.

**Weaknesses:**

- Theorem 1: The theoretical model assumes that they share the same activation on certain representations, abstracted through a single random variable $Z$. This is a strong assumption as the activation values for both privacy and fairness data might correlate but almost unlikely to lead to the same value. Does the same theoretical result hold if one consider the model that considered two different yet correlated representations $Z_1$ and $Z_2$ respectively for privacy and fairness data, i.e., $I(Z_1;Z_2)>0$?

- Proposition 1: The proposition statement seems problematic. In inequality (2), the terms inside mutual information is an expectation, which has no randomness.

- The paper discusses an interesting finding yet it will be great to further examine this finding in different setups. E.g., is it possible to improve privacy and fairness simultaneously by simply balancing the ratio between privacy and fairness data? How do privacy and fairness vary with different number of samples in SFT?

**Questions:**

See above.

---

> ### Author Response · Authors · 2024-11-23
> **Response to Reviewer Wx33**
>
> Thank you for your great efforts on the review and constructive comments. We will try our best to answer all your questions. **If you are not satisfied with our answers or have more questions, please let us know as soon as possible, so that we can try our best to answer any further questions before the deadline.**
>
> ---
>
> **Q1**: "Theorem 1: The theoretical model assumes that they share the same activation on certain representations, abstracted through a single random variable Z. This is a strong assumption as the activation values for both privacy and fairness data might correlate but almost unlikely to lead to the same value. Does the same theoretical result hold if one consider the model that considered two different yet correlated representations Z1 and Z2 respectively for privacy and fairness data, i.e., I(Z1;Z2)>0? "
>
> **A1**: Thank you for your constructive comments. We have **followed your suggestions** to derive the theoretical results under the relaxed assumption, and find that **the same theoretical result still holds**. The updated theoretical results and corresponding proofs have been added to **Appendix B of the updated manuscript**, highlighted in blue for clarity. We believe that the updated theoretical section reasonably relaxes the assumptions, thereby broadening its applicability.
>
> ---
>
> **Q2**: "The proposition 1 statement seems problematic. In inequality (2), the terms inside mutual information is an expectation, which has no randomness."
>
> **A2**: Thank you for pointing this out. In the updated version of the paper, we have revised Proposition 1 by removing the expectation to ensure that the terms inside the mutual information align with the definition of random variables.
>
> ---
>
> **Q3**: "The paper discusses an interesting finding yet it will be great to further examine this finding in different setups. E.g., is it possible to improve privacy and fairness simultaneously by simply balancing the ratio between privacy and fairness data? How do privacy and fairness vary with different number of samples in SFT?"
>
> **A3**: Thank you for your valuable suggestions. We have **followed your suggestions to conduct new experiments** to examine how the performance of privacy and fairness varies with changes in the ratio between privacy and fairness data. Specifically, using Mistral-7B-IT-v0.2 as an example, we keep the total number of FFT training samples constant with **different ratio of fairness data in the training set**. The results are summarized as follows:
>
> | **Ratio of fairness data** | origin | 30%    | 40%    | 50%    | 60%    | 70%    | 80%    | 90%    | 100%   |
> | -------------------------- | ------ | ------ | ------ | ------ | ------ | ------ | ------ | ------ | ------ |
> | **Awareness of fairness**  | 0.6231 | 0.5991 | 0.6199 | 0.5570 | 0.5949 | 0.5737 | 0.5487 | 0.5926 | 0.6245 |
> | **Awareness of privacy**   | 0.6636 | 0.7839 | 0.8158 | 0.7793 | 0.7884 | 0.7549 | 0.7458 | 0.7489 | 0.347  |
>
> The experimental results indicate that:
>
> - **Simply balancing the ratio between privacy and fairness data can not improve privacy and fairness simultaneously**. When the ratio of fairness data is less than 100%, the model's fairness awareness consistently degrades, and privacy awareness consistently improves. When the training data consists entirely of fairness data, there is a slight improvement in fairness awareness, while model's privacy awareness degrades significantly.
> - **The relationship between fairness and privacy performance does not show a clear trend as the ratio of fairness data changes.** We hypothesize that other underlying factors, such as whether the LLM has encountered similar SFT data during pretraining, may affect the results.

---

> ### Author Response · Authors · 2024-11-29
> **A Gentle Reminder**
>
> Dear Reviewer Wx33,
>
> We sincerely appreciate you for your time and efforts in reviewing our paper.
>
> As the end of the discussion period is approaching, we are eagerly anticipating your feedback to ascertain if our responses have adequately addressed your concerns. Your feedback will be instrumental in enhancing the quality of our manuscript. We would appreciate the opportunity to address any additional concerns you may have before the discussion period ends.
>
> Thank you once again for your time and consideration.
>
> Sincerely,
>
> Submission621 Authors

---

> > ### Comment · Reviewer_Wx33 · 2024-12-02
> > **Thank you for the responses!**
> >
> > I appreciate the authors' efforts to improve the paper.
> >
> > > Proposition 1.
> >
> > The proposition is problematic because the feature extractors are left undefined in the proposition statement. Please add additional details about how to apply Theorem 1 to obtain Proposition 1.
> >
> > > The additional experimental results
> >
> > The results seem to conflict with the main argument of the paper that SFT enhances privacy at the cost of fairness. Increasing the data for SFT from 30% to 40% simultaneously increases privacy and fairness.
> >
> > Given these issues, I would recommend the paper to be thoroughly revised for the next cycle.

---

### Note · Authors · 2024-12-16

I have read and agree with the venue's withdrawal policy on behalf of myself and my co-authors.